Plasticity of gene expression according to salinity in the testis of broodstock and F1 black-chinned tilapia, Sarotherodon melanotheron heudelotii

Avarre Jean-Christophe 1 jean-christophe.avarre@ird.fr
Guinand Bruno 1
Dugué Rémi 1
Cosson Jacky 2
Legendre Marc 1
Panfili Jacques 3
Durand Jean-Dominique 3
1 Institut des Sciences de l’Evolution de Montpellier , UMR 226 IRD-CNRS-UM2, Montpellier , France
2 Faculty of Fisheries and Protection of Waters, South Bohemian Research Center of Aquaculture and Biodiversity of Hydrocenoses, Research Institute of Fish Culture and Hydrobiology, University of South Bohemia in Ceske Budejovice , Vodňany , Czech Republic
3 Ecologie des Systèmes Marins Côtiers , UMR 5119 IRD-UM2-CNRS-IFREMER, Montpellier , France
Esteban María Ángeles
Electronic publication date: 2014 Dec 18
Publication date: 2014
Volume: 2
Electronic Location ID: e702
Received 2014 Oct 20; Accepted 2014 Nov 27
Copyright: © 2014 Avarre et al.
Copyright year: 2014
Copyright holder: Avarre et al.
License: This is an open access article distributed under the terms of the Creative Commons Attribution License, which permits unrestricted use, distribution, reproduction and adaptation in any medium and for any purpose provided that it is properly attributed. For attribution, the original author(s), title, publication source (PeerJ) and either DOI or URL of the article must be cited.
License URL: https://creativecommons.org/licenses/by/4.0/

Keywords: Male reproduction, Salinity, Gene expression, Acclimation, Fish

Funding: INSU (Institut National des Sciences de l’Univers)—EC2CO (Ecosphère Continentale et Côtière) (2010-2012) GACR P502/12/1973 LD14119 CENAKVA CZ.1.05/2.1.00/01.0024 LO1205 National CZ COST project LD14119 This study was supported by an INSU (Institut National des Sciences de l’Univers)—EC2CO (Ecosphère Continentale et Côtière) grant (2010-2012). Jean-Christophe Avarre received funding from GACR P502/12/1973, LD14119, CENAKVA CZ.1.05/2.1.00/01.0024 and LO1205 and also from the national CZ COST project as part of the International COST action “Aquagamete”, number LD14119. The results of the project LO1205 were obtained with financial support from the MEYS of the CR under the NPU I program. The funders had no role in study design, data collection and analysis, decision to publish, or preparation of the manuscript.

==============================
The black-chinned tilapia Sarotherodon melanotheron heudelotii Rüppell 1852 (Teleostei, Cichlidae) displays remarkable acclimation capacities. When exposed to drastic changes of salinity, which can be the case in its natural habitat, it develops quick physiological responses and keeps reproducing. The present study focused on the physiological impact of salinity on male reproductive capacities, using gene expression as a proxy of acclimation process. Two series of experimental fish were investigated: the first one was composed of fish maintained in freshwater for several generations and newly acclimated to salinities of 35 and 70, whereas the second one consisted of the descendants of the latter born and were raised under their native salinity. Expression patterns of 43 candidate genes previously identified from the testes of wild males was investigated in the three salinities and two generations. Twenty of them showed significant expression differences between salinities, and their predicted function revealed that most of them are involved in the osmotic tolerance of sperm cells and/or in the maintenance of sperm motility. A high level of expression variation was evidenced, especially for fish maintained in freshwater. In spite of this, gene expression patterns allowed the differentiation between fish raised in freshwater and those maintained in hypersaline water in both generations. Altogether, the results presented here suggest that this high variability of expression is likely to ensure the reproductive success of this species under varying salinities.

Introduction

The black-chinned tilapia, Sarotherodon melanotheron heudelotii Rüppell 1852 (Teleostei, Cichlidae), is a mouth-brooding fish that mainly occurs in estuarine and lagoon ecosystems of West Africa, but also sometimes in isolated, natural or artificial ponds. This species is an important local fish resource, accounting for a large part of catches in this area. Because of reduced freshwater input and the intense evaporation that has occurred over the last years (Pagès & Citeau, 1990; Savenije & Pages, 1992), it is regularly exposed to changes of salinity in its natural habitats. This led to physiological modifications of osmoregulation (Lorin-Nebel et al., 2012; Tine et al., 2011) and reproductive strategies (Gueye et al., 2012; Legendre et al., 2008; Panfili et al., 2004; Panfili et al., 2006).

Analysis of the gene expression patterns in the gills of this species evidenced a clear differentiation of sub-populations along the Sine Saloum estuary (Senegal) linked to the ambient salinity (Tine, Guinand & Durand, 2012). It was also demonstrated that salinity induced phenotypic modifications of the mechanisms involved in the activation of sperm cell motility, one of the major indicators of male fitness (Fauvel, Suquet & Cosson, 2010; Lahnsteiner et al., 1998). Indeed, the osmolality that enabled sperm activation in the black-chinned tilapia increased significantly with the salinity at which broodfish were maintained (Legendre et al., 2008). This finding was also recently reported for another estuarine species, Fundulus grandis (Tiersch & Yang, 2012), indicating that increased knowledge in S. m. heudelotii could also benefit other euryhaline species.

Most of the studies focusing on the response of aquatic animals to alternative osmotic (Evans & Somero, 2008; Larsen et al., 2007; Whitehead & Crawford, 2006; Whitehead et al., 2011) or hypoxic (Gracey, 2007; Rathburn et al., 2013; Tiedke, Thiel & Burmester, 2014) environments addressed this question through functional genomics of the gills and/or liver. Nevertheless, the fitness of individuals does not only rely on short-term, direct physiological responses to environmental challenges, but also on their capacity to produce viable gametes and offspring under a wide range of environmental conditions (Breckels & Neff, 2013; Dorts et al., 2012). Salinity has been shown to significantly modify some reproductive traits such as length at first sexual maturity, fecundity and oocyte size in the wild (Diouf et al., 2009; Panfili et al., 2004; Panfili et al., 2006; Whiterod & Walker, 2006). However, very few studies attempted to investigate how gene expression in gonads responded to salinity challenges, even though gametogenesis and gamete quality may be highly influenced by salinity (Alavi & Cosson, 2006; Bobe & Labbé, 2010; Cosson, 2004).

Using a high-throughput transcriptomic approach, Avarre et al. (2014) validated a de novo qPCR assay complying with the MIQE (Minimum Information for publication of Quantitative real-time PCR Experiments) guidelines (Bustin et al., 2009) for 43 candidate and 11 reference gene transcripts in the testes of mature males sampled in Senegal at locations displaying salinities of 40 and 95. The aim of the present study was to examine the putative involvement of the expression pattern of these 43 candidate genes in the acclimation of male reproductive capacities to salinity changes over two generations. More specifically, it investigated (i) how transfer from low salinities (0) to high salinities (35 and 70) induced specific changes of gene expression in testes within a single parental generation, and (ii) if these variations persisted in the next generation (F1). Two series of experimental fish were analyzed: the first one was composed of fish maintained in freshwater and newly acclimated to salinities of 35 and 70 (“transferred fish”, T), whereas the second one consisted of the F1 descendants of T fish. These F1 individuals were born and raised in each salinity condition (“born fish”, B). Levels of gene expression were compared between the two generations in order to bring insights into the mechanisms that allow S. melanotheron males to respond to salinity changes without compromising the success of reproduction.

Material and Methods

Fish samples

The fish used in this study came from a single freshwater strain of Sarotherodon melanotheron heudelotii. They originated from a population of ∼50 juveniles sampled in the Niayes (natural freshwater ponds) of Dakar (Senegal) and transferred to our facilities (Montpellier) nearly 15 years ago. Since then, fish have been reared in freshwater recirculation systems (i.e., in the same salinity as that of their natural environment). In order to minimize the loss of genetic diversity, 3–10 mate pairs were used to obtain a new generation, and one generation corresponded to approximately 18 months. Twelve mature males and twelve mature females (approximately 18 month old) from this pool (hereafter referred to as T) were transferred to 3 independent water recirculation systems dedicated to a specific salinity (i.e., 0, 35 and 70). These salinities mimic fresh, saline and hypersaline waters in which S. melanotheron heudelotii may be frequently encountered in the wild. Because of technical limitations, experiments with higher water salinities were not implemented here. Each system comprised two breeding tanks (polyester tanks of 2.5 m length × 0.53 m width × 0.30 m depth). Water salinity was gradually increased at a rate of about 1 day-1 by the addition of synthetic sea salt (“Instant Ocean”, Aquarium system, Sarrebourg, France) until the target salinities of 35 and 70 were reached. Following a 5-week period of acclimation at the final salinities, fish were monitored for their reproductive behaviour over 18 weeks. Specifically, male reproductive success was assessed through the number of incubating males per tank (related to the number of available couples), the effective fertilization of incubated eggs and the viability of progenies. Finally, they were anaesthetized (Eugenol, 0.1 mL/L) and then killed by an overdose of anaesthetic (Eugenol, 0.5 mL/L) in ice (in accordance with the EU Directive 2010/63/EU) for dissection and testis collection. The body weight (Wb), fork length (FL) and gonad weight (Wg) were recorded for each fish, and the gonadosomatic index (GSI) was calculated as follow: Wg/Wb × 100. Condition factor K was also calculated, according to the standard formula 100 × Wb/FL3. During this period, about 50 hatchlings from 6–8 spawning events per salinity were transferred to another series of tanks. They were grown at salinities of 0, 35 and 70 until they were 9–11 months old, i.e., about 3 months after they became sexually mature. From these large pools of animals (referred to as B), 12 males and 12 females per salinity condition were randomly selected and transferred to the same breeding tanks used for T animals. Again, their reproductive activity was monitored for 18 consecutive weeks before they were processed for testis collection as indicated above. Testes were placed in RNA later (Ambion) overnight at 4 °C and then stored at −20 °C until use. In total, testes from 33 T and 35 B fish were collected (Fig. 1). All the experimental procedures took place in our facilities in Montpellier, under the laboratory agreement for animal experimentation number A-34-172-24 and the author’s personal authorization for animal experimentation number 34-188, both provided by the French government.

Figure 1 Origin of the 2 series of experimental animals considered in this study, which both originated from a single stock maintained in captivity in freshwater since ∼15 years.

T animals result from a transfer from fish of this stock to the same (0) or to different salinity conditions (35 and 70). B animals represent first-generation male offspring from the reproduction of each category of T fish under their respective salinity conditions (0, 35 or 70). Number of T and B males collected from each salinity condition is reported in brackets.

RNA extraction and cDNA synthesis

RNA was extracted with the Nucleospin-8 total RNA isolation kit (Macherey-Nagel). Fifteen to twenty mg of testis preserved in RNA later were weighed and transferred into 2 ml tubes containing a 5 mm steel bead (Qiagen) as well as 360 µL lysis buffer supplemented with 1% β-mercaptoethanol (Sigma-Aldrich). Tissues were homogenized with a tissue lyzer (Qiagen) for 2 min at 50 Hz. Tubes were then centrifuged for 5 min at full speed and the supernatants were transferred to new tubes and kept at −20 °C overnight. RNA was extracted the following day according to the manufacturer’s instructions, using a Janus automated workstation (Perkin Elmer), and eluted in 70 µL RNase-free H2O. In order to remove any trace of contaminating genomic DNA, RNA eluates were subjected to a second DNase treatment: a mix of 0.2 µL of RNase-free DNase and 2 µL of reaction buffer (Macherey-Nagel) was added to 20 µL of each RNA eluate, and digestion was carried out for 15 min at 37 °C. RNA quantity was measured by UV spectrophotometry (Nanodrop 1000; Thermoscientific), and its integrity was verified by capillary electrophoresis (Agilent Bioanalyzer 2100). Each RNA sample was diluted to a concentration of 50 ng µL-1 in H2O and stored at −80 °C.

Reverse transcription was performed with oligodT primers on 250 µg RNA, using the transcriptor first strand cDNA synthesis kit (Roche). A template-primer mixture consisting of 250 µg RNA and 2.5 µM oligodT was denatured at 65 °C for 10-min and immediately cooled on ice. The reaction (in 20 µL final) was supplemented with reaction buffer (1X), dNTPs (1 mM each), RNase inhibitor (20 U) and reverse transcriptase (10 U), incubated for 1 h at 50 °C, then heated for 5 min at 85 °C and immediately cooled on ice. The resulting cDNAs were diluted 10 times with 180 µL H2O and stored at −20 °C until use.

Gene sequence annotation

The candidate and reference genes investigated here were identified using a high-throughput digital gene expression approach (Avarre et al., 2014). Their raw sequences can be found under SRA study accession number SRP022935, whereas the assembled sequences can be accessed through a freely accessible interactive database (http://vmdiva-proto.ird.fr). Annotation of the corresponding sequences was therefore needed to infer their putative functions. This was realized with Blast2GO v2.6.6 (Conesa et al., 2005). Sequences were used as a query to search the non-redundant protein database available at the National Center for Biotechnology Information (www.ncbi.nlm.nih.gov) using the BlastX algorithm with an E-value cutoff set at 10-6. Sequences were then functionally annotated by mapping against gene ontology (GO) resources. Sequences that were not assigned any GO term were checked for conserved domains using the CD-search tool (Marchler-Bauer & Bryant, 2004). Likewise, sequences for which the number of BlastX hits was <5 were re-aligned using the BlastN algorithm, and their description was corrected when necessary.

Gene expression analysis

The expression of 43 candidate genes previously validated for their potential as being involved in testis response to salinity was analyzed by qPCR at the 3 salinities and for the 2 fish generations. PCR amplifications were carried out in 384-well plates with a LightCycler 480 (Roche) in a final volume of 6 µL containing 3 µL of SYBR Green I Master mix (Roche), 2 µL of cDNA and 0.5 µM of each primer (Avarre et al., 2014). Amplifications were performed in duplicate with an initial denaturation step of 10 min at 95 °C followed by 40 cycles of denaturation at 95 °C for 10 s, annealing at 60 °C for 10 s and elongation at 72 °C for 10 s. Amplifications were followed by a melting procedure, consisting of a brief denaturation at 95 °C for 5 s, a cooling step at 65 °C for 1 min and a slow denaturation to 97 °C. Amplification products were validated by analyzing the shape of their corresponding melting curve and by measuring their size on agarose gel electrophoresis. For each given sample, all the genes were amplified simultaneously in the same 384-well plate, and each plate contained a no-template control for every primer pair. Cycle of quantification (Cq) values were calculated with the LightCycler software, using the second derivative method. Results were expressed as changes in relative expression according to the 2−ΔΔCq method (Pfaffl, 2001). Cq values were first corrected with the amplification efficiency of each primer pair according to the following equation: CqE=100% = CqE (log(1 + E)/log(2)), where E is the efficiency and CqE the uncorrected Cq values. Then the corrected Cqs of each gene of interest were normalized (ΔCq) with the mean Cq of 4 validated reference genes (Avarre et al., 2014), and ΔCq values were related to the average ΔCq value of all samples.

Statistical analyses

T and B animals maintained in freshwater were initially analyzed as 2 different groups; however, because they belonged to the same salinity treatment and were kept in the same conditions, they were also considered as one single group for statistical purposes. Comparison of the two situations showed that variations in gene expression followed the same trend, indicating that pooling did not mislead interpretation (not shown). All statistical analyses were performed with the GenEx Pro package (MultiD analyses, Sweden). The normality of data distribution was first verified for each series of samples using the Kolmogorov–Smirnov test. Since more than 90% of series turned out to be normally distributed, a one-way ANOVA test with a Tukey–Kramer’s post-test was applied to infer significant differences between salinities, using a confidence level of 0.95 (p < 0.05). P-values were corrected for multiple testing using the false discovery rate (Benjamini & Hochberg, 1995). Concurrently, expression levels were also compared by salinity pairs with a t-test, using the same confidence level. Finally, a principal component analysis (PCA) was also carried out on the two series of fish, according to the expression pattern of the investigated genes.

Results

In spite of their age difference, average GSI for the two series of fish (T and B) were comparable, with mean (± SD) values of 0.27 ± 0.13 and 0.26 ± 0.12 for T and B males, respectively. Condition factor calculated for each salinity group showed similar values, ranging between 1.99 ± 0.11 and 2.04 ± 0.14. Moreover, spermatozoa produced by the fish investigated in this study led to successful fertilization with viable offspring in all experimental conditions, indicating that salinity did not impair the ability of males to successfully reproduce.

Differences in gene expression between salinities and generations

A collective analysis of raw Cq values for the 43 candidate and 4 reference genes in the 33 T and 35 B fish with geNorm (Vandesompele et al., 2002) and NormFinder (Andersen, Jensen & Ørntoft, 2004) software indicated that Contig_Tilapia_90_13722 (R1), Contig_Tilapia_90_7452 (R2), Contig_Tilapia_90_3058 (R3) and Transcript_AVA3_453 (R4) were the most stably expressed genes. This confirmed that these four genes were appropriate to use as reference in the present conditions, as was already demonstrated on wild fish (Avarre et al., 2014).

Among the 43 tested candidate genes, 20 showed significant variations between fish kept at different salinities in at least one of the 2 generations investigated in this study (Table 1). The number of genes that showed significant variations in their expression levels between salinities was higher in T (18) than in B (10) animals. The relative expression levels of these genes are displayed in Figs. 2 and 3. Generally, fold-changes in relative expression between the different salinity conditions were quite low, as the highest ratio was 3.86. Conversely, inter-individual variations among salinity groups were rather high. Interestingly, these variations were uppermost in fish kept in fresh water. Within each generation, the largest differences were observed between the most extreme salinities, i.e., between 0 and 70, and to a lesser extent between 0 and 35, as indicated by Tukey–Kramer pairwise comparisons. In T fish, the number of genes showing significant differences was 11 between fresh and seawater, 15 between fresh and hypersaline water and 2 between saline and hypersaline water. In B fish, these numbers amounted to 5, 11 and 2, respectively. Likewise, the largest fold-change differences were also observed between salinities 0 and 70 for the two series of animals. Overall, directions of expression differences between salinities were comparable in both generations.

Figure 2 Expression variations according to salinity of the 20 significant genes for T animals.

Values are expressed as relative expression ± SD. Original sequence names may be found in Table 1. Bars are colored according to the salinity condition (red, freshwater; green, seawater; blue, hypersaline water). Identical letters indicate no significant differences (according to a Tukey–Kramer’s post-ANOVA test) between salinities.

Figure 3 Expression variations according to salinity of the 20 significant genes for B animals.

Values are expressed as relative expression ± SD. Original sequence names may be found in Table 1. Bars are colored according to the salinity condition (red, freshwater; green, seawater; blue, hypersaline water). Identical letters indicate no significant differences (according to a Tukey–Kramer’s post-ANOVA test) between salinities.

Table 1 List of genes showing significant differential expression between salinities in the two fish generations (T and B) and corresponding statistical values.

Original sequence namea	Gene #b	Corrected ANOVA p-values	
		T animals	B animals	
Contig_Tilapia_90_6346	1	1.34E–03	3.12E–05	
Contig_Tilapia_90_8891	2	8.76E–03	7.60E–02	
Contig_Tilapia_90_947	3	1.46E–01	6.70E–04	
Contig_Tilapia_90_6938	4	1.29E–02	1.97E–03	
Contig_Tilapia_90_21432	5	3.84E–01	3.17E–03	
Contig_Tilapia_90_1393	6	7.53E–05	7.67E–01	
Contig_Tilapia_90_10643	7	3.13E–02	1.06E–03	
Transcript_AVA3_33497	8	1.28E–05	1.60E–03	
Transcript_AVA1_24409	9	2.02E–04	1.75E–02	
Contig_Tilapia_90_26617	10	1.29E–02	1.03E–02	
Contig_Tilapia_90_2414	11	3.38E–02	3.35E–01	
Contig_Tilapia_90_2253	12	1.29E–02	1.08E–01	
Contig_Tilapia_90_2777	13	1.29E–02	9.62E–01	
Contig_Tilapia_90_8343	14	1.05E–02	9.73E–01	
Transcript_AVA1_58357	15	3.38E–02	5.67E–01	
Contig_Tilapia_90_26561	16	1.34E–03	1.07E–02	
Contig_Tilapia_90_27008	17	7.76E–04	1.08E–01	
Contig_Tilapia_90_1736	18	1.32E–02	1.08E–01	
Contig_Tilapia_90_7359	19	1.54E–04	2.65E–02	
Contig_Tilapia_90_2321	20	7.53E–05	3.35E–01	
Notes.

a Names of the sequences as they appear at http://vmdiva-proto.ird.fr.

b This gene numbering is used in Table 2, Figs. 2 and 3 and throughout the text in order to facilitate reading.

Results of the PCA, based on the expression pattern of the 20 investigated genes, are displayed in Fig. 4. The two first PC axes accounted for 87.20% and 5.40% of variation. Because the variation explained by the second axis accounted for approximately 17 times less than the first one, an ANOVA with a Tukey–Kramer’s pairwise comparison was performed on the PC scores of the first axis only. It revealed that freshwater males could be significantly differentiated from T35 (P = 0.024), T70 (P = 0.0015) and B70 (P = 0.0004) animals, but not from B35 ones (P > 0.05). Moreover, there were no significant differences between T35 and B35 animals or between T70 and B70 animals (P > 0.05).

Figure 4 Principal component analysis of the fish groups according to their gene expression pattern.

The first axis accounted for 87.20% of the total variance and the second axis for 5.40%. The ellipses include 95% of the variance within each group, and the stars represent the gravity center of each ellipse. According to a Tukey–Kramer’s pairwise comparison test performed on the principal component scores of the first axis, freshwater animals (group 0) are significantly differentiated from those of groups T35 (P = 0.024), T70 (P = 0.0015) and B70 (P = 0.0004).

Predicted function of the differentially expressed and reference genes

Except for Contig_Tilapia_90_2321 which returned no significant blastX hit and Transcript_AVA3_33497 which matched a hypothetical protein, all 24 genes could be attributed to either a known or a predicted function with rather low E-values. Among these, 21 could be assigned at least one GO description (Table 2). The list contains proteins for which a putative role in spermatogenesis has already been proven in other organisms (MORC family CW-type zinc finger 2 protein, 28 kDa heat- and acid-stable phosphoprotein, seminal plasma glycoprotein), as well as proteins involved in energy metabolism (NADH dehydrogenases, phosphatase), in stress response and osmoregulation (heat shock proteins, sodium potassium ATPase), or in axonemal activity (calcium-binding protein, beta-tubulin). The predicted function of the 23 remaining genes, which did not show any significant variations according to salinity, is displayed in Table S1.

Table 2 Annotation features of the 20 responsive and 4 reference genes.

Gene #	Sequence
length (nt)	#BlastX
hitsa	Protein
descriptionb	Best hit
accession	E-Value	#GO terms	
Differentially expressed genes	
1	353	16	PREDICTED: type-4 ice-structuring
protein-like	XP_004549213	1.06533E–61	2	
2	584	20	PREDICTED: tubulin-specific
chaperone A-like	XP_003455155	4.68328E–50	5	
3	524	20	sodium/potassium-transporting
ATPase alpha-1 subunit	AGO02179	4.12826E–87	20	
4	289	20	Serine/threonine-protein
phosphatase 6 catalytic subunit	ELW48549	1.44788E–46	6	
5	164	20	PREDICTED: sperm plasma
glycoprotein 120	XP_004574936	4.12231E–23	0	
6	849	20	PREDICTED: proteasome
subunit alpha type-5-like	XP_003441568	8.42686E–175	19	
7	459	20	PREDICTED: neuroserpin-like	XP_004561427	1.13288E–76	4	
8	177	1	Hypothetical protein	XP_004563130	5.19086E–18	0	
9	260	20	PREDICTED: NADH dehydrogenase
[ubiquinone] 1 beta subcomplex
subunit 3-like	XP_003457472	8.68852E–26	2	
10	179	20	NADH dehydrogenase subunit 1	ADR10264	3.32857E–21	4	
11	604	20	PREDICTED: hypoxia-induced gene
domain family member 1A-like	XP_003438136	4.44514E–47	1	
12	545	20	Heat shock protein 90	CAX33858	8.51839E–92	34	
13	1,010	20	Heat shock protein 70	ACI42865	0.0	3	
14	506	20	PREDICTED: glutathione
S-transferase theta-1-like	XP_004572434	1.57519E–28	1	
15	321	20	PREDICTED: calcium-binding
protein 39-like isoform X1	XP_004573664	6.54477E–61	8	
16	230	20	Beta tubulin	BAD11697	7.71535E–48	7	
17	313	20	PREDICTED: MORC family CW-type
zinc finger protein 2A-like	XP_004544577	5.08988E–31	2	
18	1,105	20	PREDICTED: nucleolar protein 56-like	XP_004545283	3.59979E–151	8	
19	465	20	PREDICTED: 28 kDa heat- and
acid-stable phosphoprotein-like	XP_003443172	4.92124E–54	4	
20	512	0	–	–	–	0	
Reference genes	
R1	291	20	PREDICTED: NADH dehydrogenase
[ubiquinone] flavoprotein 1,
mitochondrial-like	XP_003452502	5.14723E–58	8	
R2	299	20	PREDICTED: NADH dehydrogenase
[ubiquinone] 1 alpha subcomplex
subunit 10, mitochondrial-like	XP_004571622	7.27748E–52	6	
R3	265	20	PREDICTED: tubulin beta-4B chain	XP_004005609	1.57111E–59	21	
R4	323	20	PREDICTED: cytochrome
c oxidase subunit 6C-1-like	XP_003451899	3.40807E–36	6	
Notes.

a The number of BlastX hits was limited to 20.

b According to the best blast hit.

Discussion

This study aimed to analyze how mature males originating from a single freshwater population responded to transfer in saline (salinity 35) or hypersaline (salinity 70) water, and how F1 individuals born in these new environments expressed the same genes. Salinity changes were shown to noticeably modify the life-histories and reproductive strategies of populations (Gueye et al., 2012; Legendre et al., 2008; Panfili et al., 2004; Panfili et al., 2006), and to impact their osmoregulatory capacities (Lorin-Nebel et al., 2012; Tine, Guinand & Durand, 2012) and their stress response (Tine et al., 2010). Nevertheless, gene expression variation in reproductive organs like testes was never investigated, although they are necessary to the preservation of male fitness and contribute to the demographic features of populations. Previous results have shown that the salinity of the water under which fish were raised had a major effect on sperm characteristics and on the conditions for the activation of spermatozoa motility. Particularly, higher osmolality and higher concentrations of extracellular calcium were required for the activation of spermatozoa in fish maintained in saline/hypersaline water (Legendre et al., 2008 and M Legendre, 2014, unpublished data). The precise mechanisms, and especially the molecular basis behind these physiological adaptations, still need further investigations. However, as intra-testicular spermatozoa and other testicular cell types are particularly difficult to separate, and the quantification of gene expression for each single cell type difficult to reach, studying gene expression variation at the testis level is a necessary step for deciphering the genes involved in salinity acclimation. Physiological changes related to environmental salinity were observed not only at the spermatozoa level (motility activation), but also at the gonad level (e.g., ionic content and osmolarity of the seminal fluid involved in the protection of spermatozoa during storage in the reproductive system) (M Legendre, 2014, unpublished data), indicating that a tissue level approach is necessary.

Global expression patterns of the 20 genes distinguished fish that only experienced freshwater from their counterparts acclimated to saline and hypersaline conditions (T35 and T70). This reflected a shift—i.e., a plastic response—in mean levels of gene expression from a standard freshwater environment to a new environment, with no significant differences in global gene expression patterns between T35 and T70 males. Among B animals (i.e., born in a specific salinity), gene expression patterns were found significantly different only between freshwater and B70 males. Inter-individual variation in mRNA levels was found highest for T and B fish maintained in freshwater, i.e., the only animals that did not undergo any environmental change for many generations. Concurrently, inter-individual variation in gene expression was around twice lower in T35, B35 and T70 males, and 3 times lower in B70 fish. Elevated inter-individual variations in mRNA levels have been proposed as a possible source of variation to enable future evolution in reaction to rapid environmental changes (Aubin-Horth et al., 2005; Oleksiak, Churchill & Crawford, 2002; Whitehead & Crawford, 2005; Whitehead & Crawford, 2006), since production of better adapted protein orthologs does not fit such short time-scales (Hofmann & Todgham, 2010). By contrast with gene expression levels, no differences in fertilization capabilities were observed between all investigated fish. If the salinity increase induced a shift in the osmolarity at which sperm cells were activated (not shown), it did not yet affect sperm motility itself, and spermatozoa produced by all the fish led to successful fertilization with viable offspring in all experimental situations. This indicates that salinity changes and salinity itself did not impair the ability of males to reproduce successfully. Combined these findings suggest that the genes investigated here are involved in the mechanisms of acclimation to salinity. They also support the hypothesis that naturally-occurring expression variation contributes to the phenotypic plasticity of male black-chinned tilapia, which ensures its reproductive success under varying salinities. Nevertheless, this plasticity may differ between T and B fish. Indeed, T fish demonstrated an ability to respond to a punctual, context-dependent change in environmental conditions after being raised in a common environment, a process known as phenotypic flexibility (Piersma & Drent, 2003). In contrast, B fish ‘accomodated’ their respective saline environment since hatching, and differences in gene expression across treatments might partly originate from the developmental component of gene expression plasticity (West-Eberhard, 2003). The present experimental design does not permit us to conclude whether the differences found between T and B animals (in terms of gene expression) are due to phenotypic flexibility itself or to the co-occurrence of both types of plasticity. Detailed studies of these two components of plasticity warrant further investigations into the black-chinned tilapia.

Among the 20 genes showing differential expression in the testes, many encode proteins that have a link with the general oxidation–reduction level of sperm cells, and/or participate to plasma membrane channel activity through differential regulation of ion content. Both activities play an important role in the osmotic tolerance of sperm cells (Morita et al., 2011) and in the maintenance of sperm motility in fish (Alavi & Cosson, 2006). The potential involvement of some of these genes is discussed below.

Among the largest variations in gene expression that were observed, gene 1 encodes a protein homologous to type-IV ice-structuring protein, also known as antifreeze protein type-IV (AFPIV). The AFPIV has already been reported in many fishes from cold, temperate and warm waters (Lee et al., 2011), and its role is still subject to question. It was indeed shown to have actual antifreeze ability. However, its low plasma level measured in the longhorn sculpin (Myoxocephalus octodecimspinosis) suggests another function for this protein, such as a role in lipid transport due to its close structure relatedness with that of fish apolipoproteins (Gauthier et al., 2008). Recently, AFPIV was shown to be abundantly synthesized in ovaries of the Atlantic cod (Gadus morhua), especially during late stages of vitellogenesis, and was proposed to be involved in lipid transport and/or metabolism (Breton et al., 2012), in spite of a proven low concentration in the blood of adults (Gauthier et al., 2008). Finally, expression of AFPIV was also demonstrated in embryos of the gibel carp (Carassius auratus gibelio), and the authors proposed a potential role in the embryonic patterning (Liu, Zhai & Gui, 2009). Beyond their role in fish, it is known that antifreeze proteins participate to the osmotic resistance of spermatozoa by reducing mechanical stress to the cell membrane. They are hence often used in sperm cryopreservation (Prathalingam et al., 2006). Genes or more general factors regulating osmolality and ion content are central to sperm motility (Cosson et al., 2008). In both T and B fish, expression of gene 1 significantly decreased with the salinity to which tilapia fish specimens were exposed. Experimental evidence is now required to elucidate the potential role of an antifreeze type IV-related protein in the adaptation to salinity, especially with regard to male gonadic activity and fertility. This is the first time that expression of an AFPIV is reported to occur in testes.

If the expression of the Na+/K+ ATPase (NKA) gene has been extensively studied in the gills of fish exposed to different salinities (Havird, Henry & Wilson, 2013), including S. melanotheron (Lorin-Nebel et al., 2012; Tine et al., 2008; Tine, Guinand & Durand, 2012), this is only the first report of its expression in the testes (gene 3). In gills, NKA plays an essential role in osmoregulation through branchial ionocytes to actively uptake/excrete ions from/to environmental water, respectively. Since testes have no ionocytes, it is likely that NKA plays other roles in the male gonads, and this requires further investigation. Sequence 4 matches a portion of the catalytic subunit of a serine-threonine protein phosphatase. A modulatory role of serine-threonine protein phosphatase in osmoregulation has been demonstrated in fish (Marshall, Ossum & Hoffmann, 2005), but details are scarce and the mechanisms still poorly explained, most likely because of the myriad reactions controlled by serine-threonine protein phosphatases (Shi, 2009). The involvement of several serine-threonine protein phosphatases in the regulation of sperm motility was recently demonstrated in humans (Fardilha et al., 2013); this research has to be extended to fish. Sequence 5 is similar to that of a seminal plasma glycoprotein that contains both a partial von Willebrand factor type D domain and a zona pellucida (ZP) domain. This high molecular weight glyco-protein was shown to have a sperm-binding activity and a sperm-immobilizing activity (Mochida et al., 2002).

The predicted function of gene 15 points to a possible calcium-binding activity. A significant raise of its abundance was observed with salinity in T animals, suggesting an increase of Ca2+ metabolism in the testes of fish exposed to higher salinities. Earlier studies on another euryhaline tilapia, Oreochromis mossambicus, suggested that acclimation of sperm motility to salinity was associated with a modulation of the Ca2+ flow in order to increase its intracellular concentration (Morita, Takemura & Okuno, 2004). More recent studies on the black-chinned tilapia showed that the osmolality that enabled sperm activation increased significantly with the salinity at which broodfish were maintained. It was also found that increasing amounts of calcium in the sperm activation medium were needed to initiate sperm motility as a function of fish rearing salinity (Legendre et al., 2008).

Sequence 16 matches a beta-tubulin, which involvement in the flagellar motility, especially through post-translational modifications, has been shown for a wide range of organisms (Huitorel et al., 1999). The description of gene 17 matches a MORC family CW-type zinc (Zn) finger protein, which absence was first shown to trigger the stop of spermatogenesis in mice (Watson et al., 1998). Zinc is a trace element essential to reproduction in both sexes of numerous mammalian species including humans (Bedwal & Bahuguna, 1994). Its involvement in spermatogenesis was recently shown in the Japanese eel (Anguilla japonica) by activating Zn-finger proteins and modulating transcription factor genes containing Zn-finger motifs (Yamaguchi et al., 2009). It allows for the regulation of mitotic cell proliferation and meiosis, the activation/inactivation of sperm motility, and may also affect the regulation of steroid hormone receptors including androgens (Yamaguchi et al., 2009).

Among the 24 (reference and differentially expressed) genes analyzed in this study, four encode predicted NADH dehydrogenase subunits: two of them showed differential expression between salinities (genes 9 and 10), whereas the other two were used as reference (R1 and R2). Expression of NADH dehydrogenase was already demonstrated to significantly vary with salinity in the gills of the black-chinned tilapia (Tine et al., 2010; Tine et al., 2008; Tine, Guinand & Durand, 2012), but transcripts were not annotated precisely at that time. The two differentially regulated transcripts found in this study correspond to subunits ND1 and NDUFB3/B12, whereas the two sequences used as reference correspond to subunits NDUFV1 and NDUFA10. All of these NADH dehydrogenase subunits are part of a complicated multiprotein complex located in the inner mitochondrial membrane, the NADH:ubiquinone oxidoreductase (complex I). It plays a central role in oxidative phosphorylation and its main function is the transport of electrons by oxidation of NADH followed by reduction of ubiquinone, which is accompanied by the translocation of protons from the mitochondrial matrix to the inter-membrane space (Loeffen et al., 1998). In humans, correlations between sperm quality or sperm motility and mitochondrial activities including oxidative phosphorylation have been known for some time (Piomboni et al., 2012; Ruiz-Pesini et al., 1998). Complex I is composed of 45 different subunits, regulated by both nuclear and mitochondrial genomes (Lazarou et al., 2009). It is therefore not surprising that different subunits may be subjected to differing regulation pathways, depending on available substrates and on the physico-chemical conditions in which they operate, as reported in humans (Piomboni et al., 2012). This was recently shown in fish by a study analyzing the transcriptional regulation during the ovarian development of the Senegalese sole (Solea senegalensis) (Tingaud-Sequeira et al., 2009), but data on male-specific tissues such as testes are lacking. Variations in complex I activity have been reported in many species, especially in the case of altered environments, and a reduction of its activity with water temperature was recently shown in Fundulus heteroclitus (Loftus & Crawford, 2013). In the present study, expression of the two above-mentioned subunits significantly decreased with salinity. This differential expression could be related to the link of complex I with reactive oxygen species (ROS) (Maranzana et al., 2013), which are known to be involved in the control of sperm motility, both in mammals (de Lamirande et al., 1997) and fish (Shaliutina et al., 2014). Indeed, a recent study pointed out a relationship between the oxidation–reduction level and the phosphorylation status of an 18-kDa superoxide anion scavenger protein in the sperm cells of Oreochromis mossambicus, and showed that ROS-dependent mechanisms contributed to the osmotic tolerance of this other euryhaline tilapia (Morita et al., 2011).

Conclusion

The present study enabled the identification of 20 candidate genes likely involved in the acclimation to salinity changes of the reproductive physiology of Sarotherodon melanotheron heudelotii. It evidenced the potential role of unexpected transcripts (such as that encoding an antifreeze protein type-IV), and supported the hypothesis that elevated variations in gene expression may contribute to the remarkable plasticity of this species. The relative contribution of phenotypic flexibility and developmental plasticity needs to be investigated in more detail, in order to gain further understanding on the fitness consequences of such changes in testis gene expression.

Supplemental Information

Table S1 Annotation features of the 23 non-responsive genes

Click here for additional data file.

We are very grateful to Mr. Christophe Cochet for his strong involvement in the maintenance of fish welfare. This is publication IRD-DIVA-ISEM-2014-190.

Additional Information and Declarations

Competing Interests

Author Contributions

Animal Ethics

Data Deposition

The authors declare there are no competing interests.

Jean-Christophe Avarre conceived and designed the experiments, performed the experiments, analyzed the data, contributed reagents/materials/analysis tools, wrote the paper, prepared figures and/or tables, reviewed drafts of the paper.

Bruno Guinand analyzed the data, wrote the paper, prepared figures and/or tables, reviewed drafts of the paper.

Rémi Dugué performed the experiments, reviewed drafts of the paper.

Jacky Cosson, Marc Legendre and Jacques Panfili performed the experiments, contributed reagents/materials/analysis tools, reviewed drafts of the paper.

Jean-Dominique Durand conceived and designed the experiments, wrote the paper, reviewed drafts of the paper.

The following information was supplied relating to ethical approvals (i.e., approving body and any reference numbers):

All the experimental procedures took place in our facilities in Montpellier, under the laboratory agreement for animal experimentation number A-34-172-24 and the author’s personal authorization for animal experimentation number 34-188, both delivered by the French government.

The following information was supplied regarding the deposition of related data:

Sequence Read Archive #SRP022935 “Sarotherodon melanotheron heudelotii RNA-seq”.

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
