# Peer review of "Plasticity of gene expression according to salinity in the testis of broodstock and F1 black-chinned tilapia, Sarotherodon melanotheron heudelotii"

_PeerJ, doi:10.7717/peerj.702_

## Round 0.1 · original submission · Minor Revisions

Dear author,
we have received the comments regarding your work.

Reviewer 1 ·

Basic reporting

This manuscript reports new and valuable insights on gene expression modulation in the testis of black-chinned tilapia in response to changes in salinity.
The paper is well written in a concise way but the result presentation is not sufficiently clear. I cannot find any logic in the order of genes in Tables, figures or text. This, together with the complex names of genes make very difficult to follow them along the MS or in the Figures. Authors must reordered the genes in Tables, Figures and text and explain the reasons of the criterion.

Experimental design

Authors have to explain why they have chosen this protocol for the salinity acclimation (rate and times). Given that they have not detected any other biological effect on the fish, they must also explain how they are sure that these conditions are representative of those occurring in natural habitats.

Obtaining accurate and reliable results from relative qRT-PCR is not an easy task; particularly in individuals maintaining their genetic diversity. As the authors explain, usually, the main problem is to find the appropriate housekeeping genes. Along the MS, authors remark the importance of inter-individual variability in gene expression as indicative of developmental plasticity in adaptive mechanisms. In this context, this variability become of great importance. So, authors should explain how they calculated error bars in Figure 1 and 2, what these bars are representing (SD, SEM, etc.). They must also show the results of the analyses mentioned in lines 179-184, in order to know which percentage of this variability in Figures 1 and 2 have to be attributed to the variations of the reference genes.

Validity of the findings

Authors state (several times) that 43 genes have been investigated, but only the 20 genes that are differentially expressed at mRNA level are showed. The knowledge of the 23 genes that are not responsive to salinity changes is also of interest. Authors should include a table with these genes and the proteins that they encode.

Please, check the last sentence about the PCA analysis (line 206). It is in contradiction with the graphic in Figure 4 and with the discussion (lines 240-244).

This paper constitutes a correct work on gene expression at transcriptional level, but the results don’t support some affirmations that are made along the text. For instance, it is too exaggerated to state repeatedly (both in the discussion and conclusions, lines 260 and 370, respectively) that the differentially expressed genes are “actively” involved in the mechanisms of acclimation to salinity changes. This sort of affirmations needs additional studies from different experimental approaches. Authors should test (e.g., by Western blot) if the differential expression a transcriptional level is correlated with differential level of the corresponding proteins, but at least they must check the MS and correct these assertions.

Reviewer 2 ·

Basic reporting

Line 237. Add correct reference in the position at [8].
Line 299-306. In gills, NKA plays an essential role in osmoregulation by branchial ionocytes to uptake/excrete ions actively from/to environmental water, respectively, and NKA roles in the testes should be different from that in the gills because testes have no ionocytes. So, it is better for this MS to rewrite the description I pointed out without comparison with branchial NKA expression patterns.
In Figs. 2 and 3, I recommend to the authors to use dots instead of commas for the decimal points in Y axis. Add the information for error bars (maybe standard deviations) in figure legends. Change the description manner for the results of statistical analyses. It is hard for readers to know the significant differences promptly from the present style of figures because alphabets are quite far from corresponding columns.
In Fig.4, change the colors for the squares representing B35 and B70 because it may become difficult to distinguish one from the other in hard copy/small form. What is 0 (triangle) stands for in this figure? There are two 0 groups (transferred and 2nd gen) in this study. If the authors express combined data of these two 0 groups as 0, please divide into T0 and B0.

Experimental design

Why the authors use fishes at different ages in transferred (18+1 months) and 2nd gen (9 to 11 months) experiments? It is known that age is one of the major influential factors for gonadal condition. Some discussions with revised Fig. 4 (including T0 and B0) will be helpful to assess the effects of age difference in this study.

Validity of the findings

This MS lacks some fundamental information for scientific validation, like gonadosomatic index, condition factors, plasma osmolality etc… These data are kinds of classical parameters, and it is widely accepted that gonadosomatic index is useful to know the maturation status of gonadal tissues. If the authors have already obtained these data, I strongly recommend showing in results or table. These kinds of information will contribute to the fruitful discussion in this MS.

---

## Round 0.2 · accepted · Accept

The revised version of your manuscript has been considered and I am pleased to accept it.